# Formation Mechanism of High-Purity Ti_2_AlN Powders under Microwave Sintering

**DOI:** 10.3390/ma13235356

**Published:** 2020-11-26

**Authors:** Weihua Chen, Jiancheng Tang, Xinghao Lin, Yunlong Ai, Nan Ye

**Affiliations:** 1School of Material Science and Engineering, Nanchang University, No. 999, Xuefu Avenue, Nanchang 330031, China; cweihua@126.com (W.C.); yenan870831@163.com (N.Y.); 2School of Materials Science and Engineering, Nanchang Hangkong University, No. 696, South Fenhe Avenue, Nanchang 330063, China; lin853439131@126.com (X.L.); ayunlong@126.com (Y.A.); 3Huaneng Fuzhou Power Plant, No. 239 Dongan, Hangcheng Street, Changle 350200, China

**Keywords:** MAX-phase Ti_2_AlN, microwave, synthesis, high-purity powder

## Abstract

In the present study, high-purity ternary-phase nitride (Ti_2_AlN) powders were synthesized through microwave sintering using TiH_2_, Al, and TiN powders as raw materials. X-ray diffraction (XRD), differential scanning calorimetry (DSC), transmission electron microscopy (TEM), and scanning electron microscopy (SEM) were adopted to characterize the as-prepared powders. It was found that the Ti_2_AlN powder prepared by the microwave sintering of the 1TiH_2_/1.15Al/1TiN mixture at 1250 °C for 30 min manifested great purity (96.68%) with uniform grain size distribution. The formation mechanism of Ti_2_AlN occurred in four stages. The solid-phase reaction of Ti/Al and Ti/TiN took place below the melting point of aluminum and formed Ti_2_Al and TiN_0.5_ phases, which were the main intermediates in Ti_2_AlN formation. Therefore, the present work puts forward a favorable method for the preparation of high-purity Ti_2_AlN powders.

## 1. Introduction

Ternary nitride and carbide compounds are novel inorganic materials that possess the properties of both metals and ceramics, such as low density, favorable chemical corrosion resistance, antioxidant capacity, simple machinability, and great electrical and thermal conductivity [1,2,3]. These ternary compounds have a general chemical formula of M_n+1_AX_n_, where M represents early transition metals, n = 1, 2, or 3, A indicates the general group of IIIA or IVA elements, and X represents either nitrogen or carbon. Similar to binary carbide rock salt structures, for every unit cell of these one-layered hexagonal lattice structures, A layers exist between the close-packed M layers, X atoms fill in M octahedral interstitial sites, and M_6_X octahedra share edges with others [4,5,6,7].

The properties of ternary carbides, such as Ti_2_AlC, Ti_3_AlC_2_, or Ti_3_SiC_2_, are widely investigated [8,9,10], and various fields, such as nuclear radiation protection [11], wear resistance [12], catalysis [13] and battery electrodes [14], have great application potential. However, the characteristics and application of ternary nitrides, such as Ti_2_AlN and Ti_4_AlN_3_, are less investigated. It is worth noting that Ti_2_AlN has almost 50% higher electrical conductivity than Ti_2_AlC [15]. In addition, the Al layers in Ti_2_AlN can be etched to the form of Ti_2_N MXene, which is a perfect potential candidate material for the energy application, especially for the use of lithium-ion batteries’ electrode materials [16]. Thus, it is of practical significance to investigate the preparation and application of Ti_2_AlN.

Ternary nitride materials are generally synthesized by different reaction pressing and sintering methods. For example, Ti_2_AlN can be prepared by hot-press sintering (HP) [17], hot isostatic pressing (HIP) [18] and spark plasma sintering (SPS) [19] of Ti/AlN or Ti/Al/TiN powders in the sintering temperature range of 1200–1400 °C. However, the aforesaid studies have mainly focused on the preparation of high-purity bulk Ti_2_AlN materials. These approaches are not appropriate for the commercial production of Ti_2_AlN powders. To conquer this problem, many pressureless sintering approaches, such as combustion synthesis (CS) [20], molten salt shielded synthesis (MS^3^) [21], mechanical alloying (MA) [22], and thermal explosion (TE) [23] are used in the preparation of Ti_2_AlN powders. Nevertheless, due to the narrow stability interval of Ti_2_AlN at high temperature, different impurity phases, such as TiAlx and TiNx, are generally formed during the above synthesis methods [24]. Therefore, it is essential to develop efficient synthesis methods to prepare high-purity Ti_2_AlN powders.

Microwave sintering (MWS) is successfully used to prepare MAX-phase ceramics because of its numerous advantages, such as rapid sintering speed, uniform heating, and lower sintering temperature [25,26,27]; however, the microwave processing for Ti_2_AlN powders is rarely reported. Therefore, the present study was conducted to examine the possibility of Ti_2_AlN preparation through MWS. All sintering processing parameters, such as raw materials, aluminum content, sintering temperature, and holding time, were optimized. Furthermore, the formation mechanism of Ti_2_AlN powders during MWS was elaborated.

## 2. Experimental Procedure

### 2.1. Materials and Methods

Commercially available titanium (Ti), titanium hydride (TiH_2_), titanium nitride (TiN), aluminum (Al), stannum (Sn), and aluminum nitride (AlN) powders were used as raw materials, and their technical features are presented in Table 1.

TiH_2_, Al, and TiN powders were first weighed and blended in a molar ratio of 1TiH_2_/(1 + x)Al/1TiN (x = 0.1, 0.15, 0.2). A suitable amount of Al was added to compensate for the Al loss due to high-temperature evaporation [28]. An appropriate amount of Sn with a molar ratio of 0.1 was also added to improve the wetting effect in the early stage of the reaction process [29,30]. In order to ensure even blending, planetary ball milling using agate milling spheres was performed at 200 rpm in an argon atmosphere for 12 h. The diameters of the agate milling spheres were 18 mm, 10 mm, and 2.5 mm with a weight ratio of 1:1:1. The ball to powder weight ratio was 5:1. The resulting slurry powder was then subjected to vacuum drying for 12 h at 60 °C. Subsequently, the mixture was filtered with a 100 mesh, and then put into an alumina crucible. Finally, microwave sintering at 480–1350 °C was followed by 1–60 min of holding. A 2.45 GHz multimode microwave sintering furnace (MWL0316V) was used for microwave sintering [31]. Argon was used as atmosphere for the sintering process, due to its inert protection and the microwave generating plasma effects [32,33]. The microwave sintering insulation device and change curves of temperature and sintering power are shown in Appendix A, respectively. Finally, the sintered sample was subjected to pulverization and grinding. For comparison, two other raw material systems, 2Ti/1AlN and 2TiH_2_/1AlN [34], were also subjected to microwave sintering for 30 min at 1250 °C.

### 2.2. Powder Characterization

A differential scanning calorimeter (DSC; 404F3, NETZSCH, Selb, Germany) was used to observe heat alterations during the sintering process. For each raw material system, 7.15 mg of the blended powder was added to the alumina crucible and heated to 1300 °C at a rate of 20 °C/min in the argon atmosphere. An X-ray diffractometer (XRD, D8ADVANCE, Bruker-AXS, Karlsruhe, Germany) was employed to identify the phase compositions of the final products under Cu-Kα radiation at 40 mA and 30 kV. The Rietveld method was employed to estimate the phase contents of the as-synthesized powders [35,36]. 

The elemental composition and microstructure of the heated specimens were analyzed by energy-dispersive spectroscopy (EDS; INCA Energy250 X-max 50, Oxford Instruments, Brno, Czech Republic) and field-emission scanning electron microscopy (FESEM, NovaNanoSEM450, FEI, Brno, Czech Republic), respectively. Moreover, the crystal topographies of the resulting Ti_2_AlN powders were analyzed by high-resolution transmission electron microscopy (HRTEM; Tecnai™ G2 F30, FEI, Brno, Czech Republic).

## 3. Results and Discussion

### 3.1. DSC Analysis

Figure 1 displays the DSC curves of the 2Ti/1AlN, 2TiH_2_/1AlN, and 1TiH_2_/1.15Al/1TiN material systems, and it is obvious that these three material systems exhibited quite distinct DSC characteristics. In the 2Ti/1AlN system (Figure 1a), only one endothermic peak was observed at about 920.8 °C. In the 2TiH_2_/1AlN system (Figure 1b), a prominent endothermic peak was observed around 555.8 °C and a weak exothermic peak was noticed at about 776.9 °C. However, in the 1TiH_2_/1.15Al/1TiN system (Figure 1c), an endothermic peak appeared around 534.5 °C and a weak endothermic peak was observed at 661.1 °C. Hence, it can be inferred that explosive thermal reactions might not occur in the 1TiH_2_/1.15Al/1TiN system with the melting reaction of aluminum. Moreover, at 893.5 °C, a weak exothermic peak was observed which could be due to the solid reaction between Ti_2_Al and TiAl_2_ compounds. 

It is noticeable that the titanium hydride-containing raw material system had a lower reaction temperature, indicating that α-Ti reacted with TiN, AlN, or Al after dehydrogenation. The dehydrogenation process might create numerous pores within the prepared powders, thus facilitating material reprocessing. Moreover, the melting process of Al favors contact between reactants. Therefore, the 1TiH_2_/1.15Al/1TiN system has the advantages of cost-effectiveness, high stability, and non-susceptibility to decomposition, thereby it could be used to synthesize high-purity Ti_2_AlN powders.

### 3.2. Factors Influencing the Synthesis of Ti_2_AlN Powders

#### 3.2.1. Raw Material Systems

The XRD patterns of the 2Ti/1AlN, 2TiH_2_/1AlN, and 1TiH_2_/1.15Al/1TiN systems prepared by microwave sintering for 30 min at 1250 °C are displayed in Figure 2. It is clear that these three material systems were composed of four phases—Ti_2_AlN (PDF#65-3496), TiN (PDF#87-0633), Ti_4_AlN_3_ (PDF#53-0444), and Ti_3_Al_2_N_2_ (PDF#80-2286). The diffraction peaks of TiN, Ti_4_AlN_3_, and Ti_3_Al_2_N_2_ were more prominent in 2Ti/1AlN and 2TiH_2_/1AlN than in the system of 1TiH_2_/1.15Al/1TiN. The estimated purity values of Ti_2_AlN powders prepared from 2Ti/1AlN, 2TiH_2_/1AlN, and 1TiH_2_/1.15Al/1TiN were 79.98, 76.78, and 96.68 wt %, respectively. Therefore, it is evident that high-purity Ti_2_AlN powder was acquired by the microwave sintering of the 1TiH_2_/1.15Al/1TiN raw material system.

#### 3.2.2. Aluminum Content

Al evaporation takes place at high temperatures, and its absence can alter the mixture proportion in a reaction system, thus a small amount of non-stoichiometric composition is necessary [28]. The XRD patterns of the samples prepared by the microwave sintering of 1TiH_2_/(1 + x)Al/1TiN (x = 0.1, 0.15, 0.2) for 30 min at 1250 °C are displayed in Figure 3. The representative diffraction peaks appeared from the Ti_2_AlN, Ti_3_Al_2_N_2_, and Ti_4_AlN_3_ phases at a TiH_2_:Al:TiN molar ratio of 1:1.1:1 (Figure 3a). When the Al molar relative content increased to 1.15, only a single-phase Ti_2_AlN ceramic with a small amount of Ti_4_AlN_3_ was obtained (Figure 3b), suggesting that the increased Al content in the mixed powder facilitated the synthesis of high-purity Ti_2_AlN as a little Al volume was possibly lost due to evaporation during sintering. At the over-weighed Al content (Figure 3c), the diffraction peaks of the Ti_4_AlN_3_ and Ti_3_Al_2_N_2_ impurity phases were enhanced. The estimated purity values of Ti_2_AlN powders prepared from 1TiH_2_/(1 + x)Al/1TiN (x = 0.1, 0.15, 0.2) were 74.85, 96.68, and 87.14 wt %, respectively. Based on these results, the value of x was set to 0.15.

#### 3.2.3. Holding Time

The XRD patterns of the 1TiH_2_/1.15Al/1TiN powder mixture sintered at 1250 °C for 10, 30, and 60 min are displayed in Figure 4. In the 1TiH_2_/1.15Al/1TiN system prepared by sintering at 1250 °C for 10 min, Ti_2_AlN existed as the primary crystalline phase with TiN, Ti_4_AlN_3_, and un-reacted TiAl (PDF#65-5414) phases (Figure 4a). However, when the holding time was extended to 30 min (Figure 4b), Ti_2_AlN and a small amount of Ti_4_AlN_3_ existed as the main phases and the diffraction peaks of TiN and TiAl almost disappeared, indicating a complete synthesis of Ti_2_AlN when sintering was carried out for 30 min at 1250 °C. When the sintering time was extended to 60 min ((Figure 4c), except for the Ti_2_AlN phase, the impurity phases such as TiN, Ti_4_AlN_3_, Ti_3_Al_2_N_2_, TiN_0.5_ (PDF#76-0198) and Al_2_O_3_ (PDF#74-1081) also existed in the system. This happened because, during this long heat preservation period, a part of the Ti_2_AlN phase was decomposed to form TiN, TiN_0.5_ and gaseous Al [37]. In addition, the gaseous Al phase was easily oxidized at 1250 °C to form a small amount of Al_2_O_3_. At the same time, the Ti_2_AlN phase tends to change into Ti_4_AlN_3_ and Ti_3_Al_2_N_2_ phases at prolonged high temperature [38]. The relative purity values of Ti_2_AlN powders prepared by sintering at 1250 °C for 10, 30, and 60 min were calculated as 84.78, 96.68, and 57.01 wt%, respectively. Therefore, the microwave sintering holding time was set as 30 min.

#### 3.2.4. Sintering Temperature

Figure 5 exhibits the XRD patterns and enlargements in the 34°–45° region of the 1TiH_2_/1.15Al/1TiN powder sample after 1 min of sintering at different temperatures (480–920 °C). It is clear from Figure 5a that at ambient temperature, the crystalline phases of the as-prepared powders were almost similar to those of TiH_2_ (PDF#25-0983), Al (PDF#85-1327), and TiN (PDF#87-0633) starting materials (except for a very small amount of Sn (PDF#65-7657)), which indicates that ball milling made no difference in the powder mixture composition. At 480 °C (Figure 5b), the phase structure did not change greatly as compared to that after ball milling. However, the peaks of 34.9° (111) and 40.5° (200) of TiH_2_ slightly shifted to higher angles, demonstrating that TiH_2_ had a tendency to dehydrogenate. When the sample was sintered at 620 °C (Figure 5c), no TiH_2_ (111) peak appeared and new TiH_1.5_ (PDF#78-2216) and α-Ti (PDF#65-3362) phases were generated from the dehydrogenation of TiH_2_ [39]. In addition, TiN_0.5_ and Ti_2_Al (PDF#47-1410) phases were also detected, suggesting that a part of TiN and Al reacted with α-Ti to form a small amount of TiN_0.5_ and Ti_2_Al under the melting point of Al. At the sintering temperature of 720 °C (Figure 5d), the peaks of TiN, Ti_2_Al, TiAl_2_ (PDF#52-0861), TiAl, and Ti_2_AlN appeared above the melting point of Al. In addition, TiH_1.5_, α-Ti, TiN_0.5_ and Al eventually disappeared, revealing the complete reaction of Al with α-Ti to Ti_2_Al, TiAl_2_, and TiAl at a sintering temperature higher than the melting point of Al. The Ti_2_AlN phase was detected at 720 °C, confirming the possibility of Ti_2_AlN synthesis at low temperatures. After microwave sintering below 820 °C (Figure 5e), no obvious change in the phase composition was noticed as compared to the sample sintered at 720 °C; however, the diffraction peaks of TiN became stronger. At 920 °C (Figure 5f), similarly to the sample prepared at 820 °C, TiN, Ti_2_Al, TiAl_2_, TiAl, and Ti_2_AlN phases were detected. However, the diffraction peaks of the TiAl and Ti_2_AlN phases were significantly enhanced. This happened because Ti_2_Al and TiAl_2_ reacted to form the TiAl phase, which then reacted with TiN to form the Ti_2_AlN phase at 920 °C.

In order to determine the optimal microwave sintering temperature, the 1TiH_2_/1.15Al/1TiN powder system was subjected to heating at different temperatures from 1150 to 1350 °C for 30 min at an interval of 50 °C, and the corresponding XRD patterns are displayed in Figure 6, where the 34°–45° region is magnified. It is clear from Figure 6a that at the sintering temperature of 1150 °C, Ti_2_AlN and TiN were the main crystalline phases with some un-reacted TiAl. However, at 1200 °C (Figure 6b), Ti_2_AlN became the dominant phase, and the diffraction peaks of TiAl and TiN almost disappeared. At the sintering temperature of 1250 °C (Figure 6c), the resulting products were composed of pure Ti_2_AlN, and the diffraction peaks of TiAl and TiN disappeared, indicating that single-phase Ti_2_AlN could be acquired at 1250 °C. At higher sintering temperatures (Figure 6d,e), the diffraction peak intensities of TiN, Ti_4_AlN_3_, TiN_0.5_, and Al_2_O_3_ increased slightly, indicating the decomposition of Ti_2_AlN to TiN formation at a sintering temperature greater than 1250 °C. The sample contained more TiN after sintering at a temperature lower than 1200 °C or higher than 1300 °C. Therefore, 1250 °C was adopted as the optimal microwave sintering temperature to synthesize high-purity Ti_2_AlN powder from the 1TiH_2_/1.15Al/1TiN raw material system.

The relative contents of TiAl, Ti_2_AlN, TiN, and Ti_4_AlN_3_ in the resultant products of the 1TiH_2_/1.15Al/1TiN system sintered for 30 min at different temperatures (1150–1350 °C) are displayed in Figure 7. It is clear that the relative content of Ti_2_AlN increased greatly and those of TiAl, Ti_2_AlC, and TiC impurity phases declined at a sintering temperature higher than 1150 °C. The relative contents of Ti_2_AlN and Ti_4_AlN_3_ at 1250 °C were calculated as 96.68 and 3.32 wt %, respectively. However, when the temperature further increased, the relative content of Ti_2_AlN gradually decreased, and those of Ti_4_AlN_3_ and TiN increased due to the transformation of Ti_2_AlN into Ti_4_AlN_3_. Therefore, 1250 °C was set as the most appropriate temperature to synthesize Ti_2_AlN.

### 3.3. Morphology and Structure of Ti_2_AlN Powders

The SEM images of the samples prepared at different sintering temperatures are displayed in Figure 8. In Figure 8a, white particles (identified as TiH_2_ (big) and TiN (small)) and gray particles (identified as Al) have a uniform distribution. When the samples were sintered at 620 °C (Figure 8b), some white particles (TiH_2_) decreased in size (1–2 µm) and acquired a lamellar shape. Moreover, the number of gray particles (Al) decreased significantly. According to the XRD result of Figure 5c, these lamellar compounds were composed of the Ti_2_Al phase. According to Figure 8c, during microwave sintering at 720 °C, more lamellar particles appeared with the disappearance of gray Al grains due to melting. It is noteworthy that these lamellar Ti–Al compounds were coated on particles. According to the XRD result of Figure 5d, lamellar Ti–Al compounds were composed of Ti_2_Al and TiAl_2_ phases, and coated particles consisted of the TiN phase. At the sintering temperature of 820 °C (Figure 8d), lamellar Ti–Al compound particles grew up on the surface of TiN, and the interface between the particles became fuzzy. This may happen due to the solid-phase diffusion reaction between Ti_2_Al and TiAl_2_ to form TiAl. At 920 °C (Figure 8e), some of these particles agglomerated and formed a layered structure. It can be speculated that TiAl and TiN reacted with each other to Ti_2_AlN at 920 °C. During sintering at 1250 °C (Figure 8f), gray Ti_2_AlN particles increased in size and had a layered morphology.

Figure 9 presents the TEM result of the 1TiH_2_/1.15Al/1TiN powder system after one minute of microwave sintering at 480 °C. 

According to the TEM image in Figure 9a and the elemental distribution mapping in Figure 9b, the distributions of Al and Ti hardly overlapped. This indicates that the distributions of TiH_2_, Al, and TiN were relatively independent at 480 °C. The phases of region A (Figure 9c) and region B (Figure 9d) corresponded to TiH_2_ and TiN, respectively. The crystal zone axis of the TiH_2_ phase was observed on [$\bar{1}$11], and the indices of crystallographic plane were (220), (202), and (422) with an interplanar distance of 0.254 nm associated with the (111) plane crystal lattice spacing. The crystal zone axis of the TiN phase was observed on [011], and the indices of the crystallographic plane were (200), ($\bar{1}\bar{1}$1), and (11$\bar{1}$) with an interplanar distance of 0.212 nm associated with the (200) plane crystal lattice spacing. This indicates that TiH_2_ and TiN had no obvious solid-phase reactions and possessed a complete structure when heated to 480 °C.

The TEM result of the 1TiH_2_/1.15Al/1TiN powder system after one minute of microwave sintering at 620 °C is exhibited in Figure 10. According to the TEM images in Figure 10a and the elemental distribution mapping in Figure 10b, the distributions party of Al and Ti overlapped each other; this indicates that they underwent solid reactions at 620 °C. The HRTEM images and selected area electron diffraction (SAED) patterns of regions C, D, and E in Figure 10a are shown in Figure 10c–e, and Figure 10f presents the EDS result of Figure 10e. It is noticeable from Figure 10c that the particles in region C possessed the TiH_1.5_ phase. The crystal zone axis of the TiH_1.5_ phase was located on [$\bar{1}$12], and the indices of the crystallographic plane were (1$\bar{1}$1), (220), and (311) with an interplanar distance of 0.252 nm associated with the (111) plane crystal lattice spacing. In the 1TiH_2_/1.15Al/1TiN system, a part of TiH_2_ was not dehydrogenated to α-Ti at 620 °C. It is noticeable from Figure 10d the particles in region D possessed the Ti_2_Al phase. The indices of the crystallographic plane of the Ti_2_Al phase were (300), (102), and (420) with an interplanar distance of 0.221 nm associated with the (211) plane crystal lattice spacing. It is observable from Figure 10e that the particles in region E had the structure of the TiN_0.5_ phase with a non-equimolar ratio of Ti and N (Figure 10f). The crystal zone axis of the TiN_0.5_ phase was located on [2$\bar{2}$0], and the indices of the crystallographic plane were (110), (002), and (112) with an interplanar distance of 0.244 nm associated with the (112) plane crystal lattice spacing. The phase boundaries of TiH_1.5_, Ti, and TiN_0.5_ were outlined based on the diverse prismatic plane spacing in each phase (Figure 10e).

Figure 11 displays the TEM results of the 1TiH_2_/1.15Al/1TiN powder system after microwave sintering at 720 and 1250 °C. It is noticeable from the TEM image in Figure 10a that each black particle was surrounded by a large number of thin flaky particles. The HRTEM image and SAED pattern of region F in Figure 11a are displayed in Figure 11b. It is clear that the flaky particles in region F possessed the TiAl_2_ phase. The indices of the crystallographic plane of the TiAl_2_ phase were (220), (200), and (116) with an interplanar distance of 0.198 nm associated with the (200) plane crystal lattice spacing. According to the XRD result of Figure 5d, when the sintering temperature was above the melting point of Al, the reaction between Ti and Al generate the flaky TiAl_2_ phase that surrounded the TiN phase (Figure 11c). The HRTEM image and SAED pattern of region G in Figure 11c are presented in Figure 11d. It is clear that region G had the Ti_2_AlN phase. The crystal zone axis of the Ti_2_AlN phase was located on [0001] and grew along with the (10$\bar{1}$0), (01$\bar{1}$0), and (11$\bar{2}$0) planes with an interplanar distance of 0.242 nm associated with the (10$\bar{1}$2) plane crystal lattice spacing. The powder sample sintered at 1250 °C possessed a hexagonal microstructure.

### 3.4. Reaction Mechanism for Ti_2_AlN Formation

The obtained data were analyzed to investigate the underlying reaction mechanism to synthesize high-purity Ti_2_AlN powder from the 1TiH_2_/1.15Al/1TiN raw material system. Table 2 summarizes the major phase components noticed after sintering at 480–1350 °C. It was found that the entire reaction process occurred in four stages.

#### 3.4.1. Reactions at Temperatures Lower than 660 °C

It is discernible from Figure 5b and Table 2 that no reaction occurred in the raw material system during sintering at 480 °C for one minute. The resulting product was composed of TiH_2_, TiN, Al, and a little amount of Sn. It can be speculated from the DSC analysis result in Figure 1 that only the Sn melting reaction occurred when the sintering temperature was lower than 480 °C [29,30]:Sn (s) → Sn (l),(1)

It is noticeable from Table 2 that the thermal dehydrogenation of TiH_2_ occurred at temperatures lower than 660 °C. The thermal dehydrogenation of TiH_2_ occurred in four steps [39,40]: TiH_1.5–2_ → TiH_1.0–1.5_ → TiH_0.2–1.0_ → TiH_0–0.2_ → Ti. Furthermore, at temperatures greater than 700 °C, the reaction was completed as follows:TiH_2_ (s) → Ti (s) + H_2_ (g),(2)

However, in the present work, the thermal dehydrogenation temperature decreased due to the two following reasons. First, Sn melting at low temperatures and mechanical ball milling led to the activation of the TiH_2_ powder surface. Second, to some extent, microwave energy can reduce the reaction temperature duo to the particularity of the heating mode [41]. 

TiN_0.5_ and flaky Ti–Al were formed at temperatures lower than the melting point of Al. The highly reactive Ti particles concentrated on Al and TiN surfaces and diffused to react with solid Al and TiN through the following solid-phase reactions: 2Ti (s) + Al (s) → Ti_2_Al (s),(3)
Ti (s) + TiN (s) → 2TiN_0.5_ (s),(4)

#### 3.4.2. Reactions at 660–720 °C

During sintering at temperatures greater than the melting point of Al (660 °C), the unmelted aluminum melted and the liquid aluminum wrapped around Ti, Ti_2_Al, TiN, and TiN_0.5_ grain surfaces. In addition, the TiN, Ti_2_Al, TiAl_2_, TiAl, and Ti_2_AlN phases were detected in the products sintered at 720 °C. The liquid Al reacted with solid Ti and Ti_2_Al to form the Al-rich Ti–Al complex. Subsequently, Al diffused from the outer surface of the Ti–Al complex to the inner Ti layer and then to the Ti_2_Al core. Moreover, Ti–Al particles were transformed into TiAl_2_ and TiAl. At 720 °C, no TiN_0.5_ phase was detected in the resultant product. However, the Ti_2_AlN phase was clearly found in the sample sintered at 720 °C, suggesting that the Ti_2_AlN phase was formed due to the reaction between liquid Al and TiN_0.5_ at 720 °C.

It is noteworthy that at 720 °C, no elemental Ti or TiH_1.5_ was found in the resultant product, this suggests that Ti and Al were completely transformed into intermediate compounds. In addition, the reaction between solid-phase particles (Ti, Ti_2_Al, TiN_0.5_) and molten Al was an exothermic one. The DSC results in Figure 1c reveals no exothermic peak at the melting point of Al; however, an endothermic peak appeared due to the melting of Al. The explosive reaction between solid and liquid phases Al was not violent at the melting point of Al; thus, facilitating the synthesis of high-purity Ti_2_AlN powder at low temperatures:Al (s) → Al (l),(5)
Ti (s) + 2Al (l) → TiAl_2_ (s),(6)
2TiN_0.5_ (s) + Al (l) → Ti_2_AlN (s),(7)
Ti_2_Al (s) + Al (l) → 2TiAl (s),(8)

#### 3.4.3. Reactions at 720–1150 °C

At the sintering temperatures of 820 and 920 °C, the TiN, TiAl, Ti_2_Al, TiAl_2_, and Ti_2_AlN phases were found in the final products. The relative contents of TiAl and Ti_2_AlN increased at 920 °C, and the diffraction peaks of Ti_2_Al and TiAl_2_ slightly decreased. However, when the temperature reached 1150 °C, no Ti_2_Al and TiAl_2_ phases were detected in the samples, and the contents of TiAl and Ti_2_AlN significantly increased. The main reaction in this stage can be speculated as
Ti_2_Al (s) + TiAl_2_ (s) → 2TiAl (s),(9)

#### 3.4.4. Reactions at 1150–1350 °C

It is evident from Table 2 that Ti_2_AlN was produced after sintering at 720 °C. Ti_2_AlN was the predominant phase when sintering was carried out at temperatures below 1250 °C, suggesting that Ti_2_AlN formation was the major reaction in this stage. The TiAl and TiN were intermediate products during the synthesis of Ti_2_AlN. TiAl and TiN were not completely transformed into Ti_2_AlN at sintering temperatures below 1200 °C and existed in the final samples: TiAl (s) + TiN (s) → Ti_2_AlN (s),(10)

When sintering was carried out at temperatures greater than 1300 °C, the TiN content greatly increased. Pang and colleagues [37,38] found that the intensity of Ti–Al bonding was lower than that of Ti–N bonding. After holding for a long time, Al was lost due to evaporation, inducing the decomposition of Ti_2_AlN. Moreover, after the loss of sufficient Al, Ti_2_AlN was transformed into TiN or TiN_0.5_ [42]. In addition, at very high temperatures, highly active gaseous Al was easily oxidized with a small amount of oxygen to form the impurity phase of Al_2_O_3_; thus, Ti_2_AlN manifested instability at temperatures higher than 1300 °C. The decomposition of Ti_2_AlN took place through the following reactions:Ti_2_AlN (s) → 2TiN(s) +Al (g) + Ti (g),(11)
Ti_2_AlN (s) → 2TiN_0.5_(s) +Al (g),(12)
Al (g) + O_2_ (g) → Al_2_O_3_ (s),(13)

It is clear that reactions (11)–(13) were the main pathways for the generation of the impurity phase at high temperatures. It should be noted that Ti_2_AlN directly reacted with TiN, accounting for another potential route for Ti_4_AlN_3_ formation [43,44]:Ti_2_AlN (s) + 2TiN (s) → Ti_4_AlN_3_ (s)(14)

## 4. Conclusions

High-purity (96.68%) Ti_2_AlN powders were successfully prepared through microwave sintering for 30 min at 1250 °C using TiH_2_, Al, and TiN powders as raw materials. In comparison to 2Ti/1AlN and 2TiH_2_/1AlN mixtures, the 1TiH_2_/1.15Al/1TiN material system manifested better performance in forming Ti_2_AlN. DSC, XRD, SEM, and TEM were conducted to examine the formation mechanism of Ti_2_AlN powders. The main observations of this research are presented below:
(1)During microwave sintering, TiH_2_ was decomposed and formed active Ti atoms at temperatures lower than the melting point of Al. Subsequently, the Ti_2_Al and TiN_0.5_ phases were formed through the reactions of active Ti atoms with Al and TiN, respectively.(2)At temperatures higher than the melting point of Al, the liquid Al easily reacted with Ti, Ti_2_Al, and TiN_0.5_ to form TiAl_2_, TiAl, and Ti_2_AlN, respectively. Subsequently, TiAl_2_ and Ti_2_Al reacted with each other to form TiAl.(3)TiAl further reacted with TiN to form Ti_2_AlN. Although Ti_2_AlN was identified as a dominant synthesized product, Ti_4_AlN_3_ did not react completely in the sintering process.

## Figures and Tables

**Figure 1 materials-13-05356-f001:**
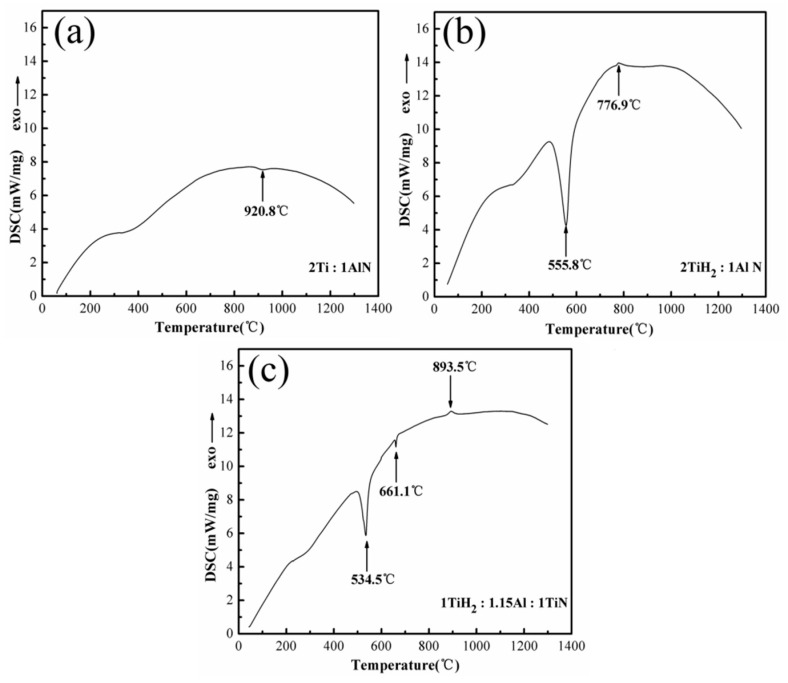
DSC curves of (**a**) 2Ti/1AlN; (**b**) 2TiH_2_/1AlN; and (**c**) 1TiH_2_/1.15Al/1TiN from 20 to 1300 °C.

**Figure 2 materials-13-05356-f002:**
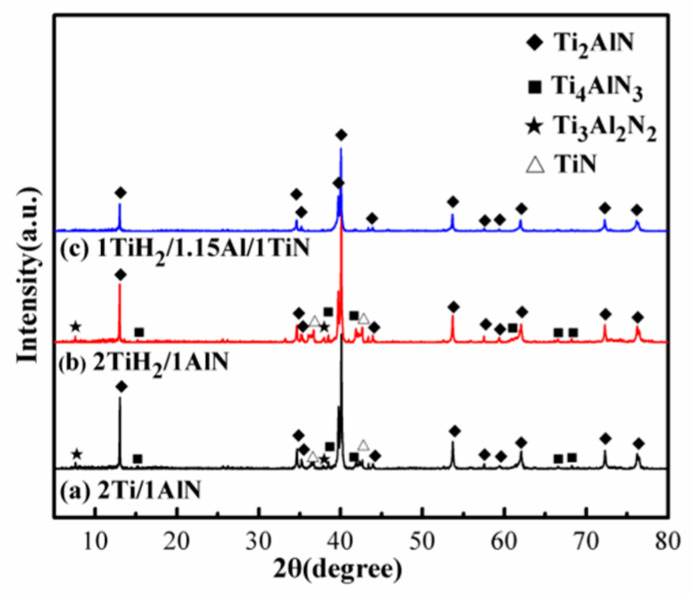
XRD patterns of (**a**) 2Ti/1AlN; (**b**) 2TiH_2_/1AlN; and (**c**) 1TiH_2_/1.15Al/1TiN prepared by microwave sintering for 30 min at 1250 °C.

**Figure 3 materials-13-05356-f003:**
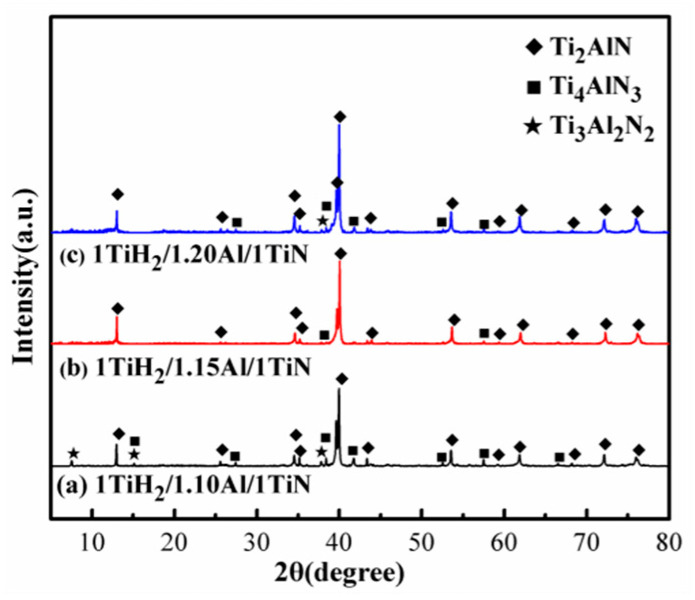
XRD patterns of the 1TiH_2_/(1 + x)Al/1TiN powder mixture prepared by microwave sintering for 30 min at 1250 °C: (**a**) x = 0.1; (**b**) x = 0.15; and (**c**) x = 0.2.

**Figure 4 materials-13-05356-f004:**
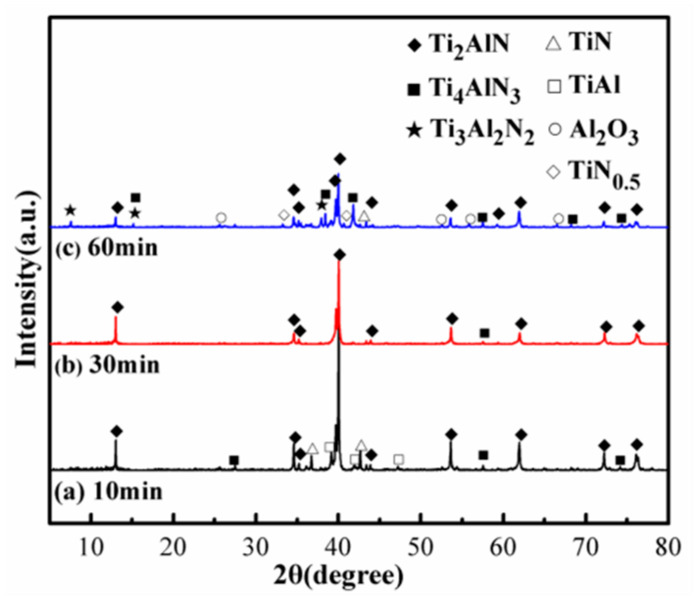
XRD patterns of the 1TiH_2_/1.15Al/TiN powder mixture sintered at 1250 °C for (**a**) 10 min; (**b**) 30 min; and (**c**) 60 min.

**Figure 5 materials-13-05356-f005:**
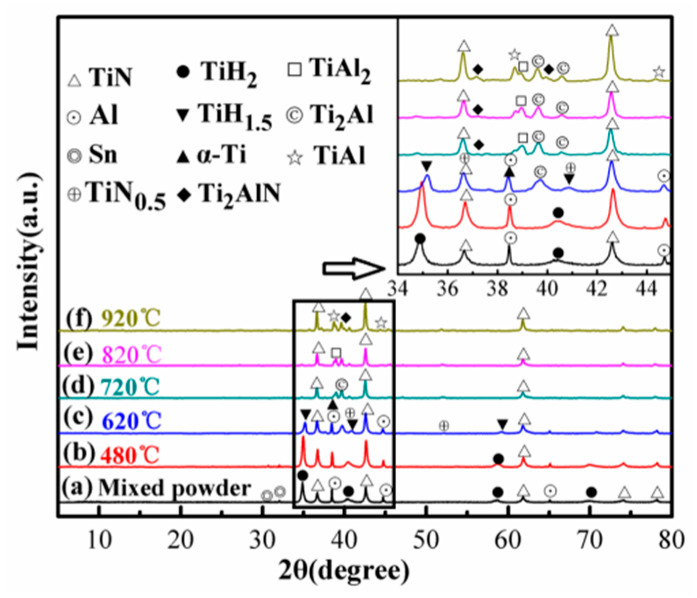
XRD patterns of the 1TiH_2_/1.15Al/1TiN powder sample after 1 min of sintering at different temperatures (480–920 °C): (**a**) Mixed powder; (**b**) 480 °C; (**c**) 620 °C; (**d**) 720 °C; (**e**) 820 °C; (**f**) 920 °C.

**Figure 6 materials-13-05356-f006:**
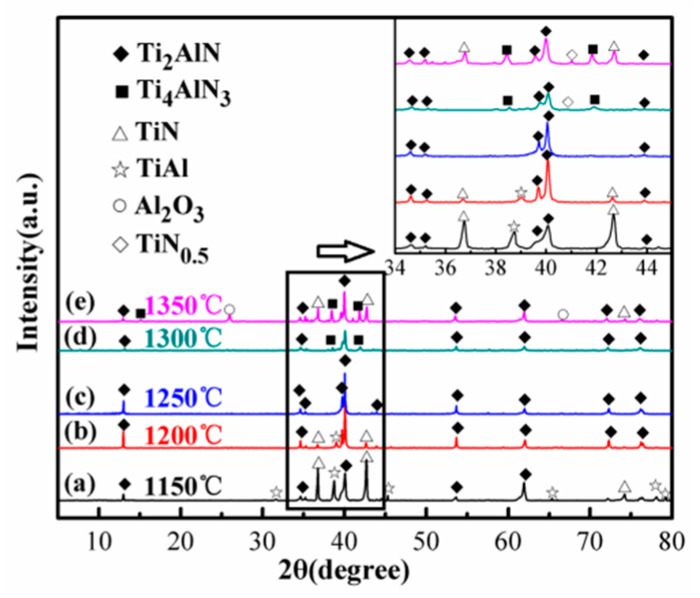
XRD patterns of the 1TiH_2_/1.15Al/1TiN powder system after 30 min of sintering at different temperatures (1150–1350 °C): (**a**) 1150 °C; (**b**) 1200 °C; (**c**) 1250 °C; (**d**) 1300 °C; (**e**) 1350 °C.

**Figure 7 materials-13-05356-f007:**
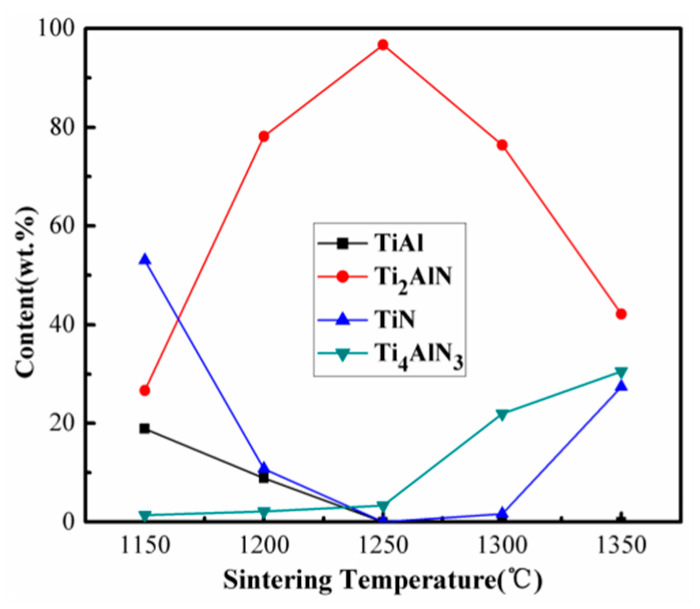
Relative contents of TiAl, Ti_2_AlN, TiN, and Ti_4_AlN_3_ in the resultant products of the 1TiH_2_/1.15Al/1TiN system after 30 min of sintering at different temperatures (1150–1350 °C).

**Figure 8 materials-13-05356-f008:**
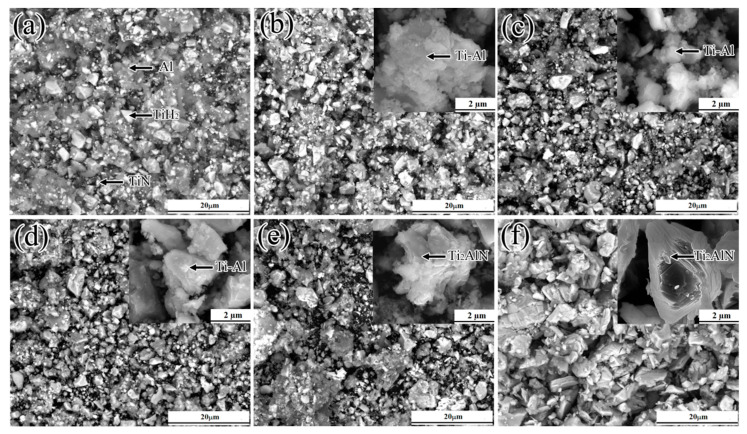
SEM images of the samples prepared at different sintering temperatures: (**a**) un-sintered 1TiH_2_/1.15Al/1TiN powder mixture; (**b**–**e**) after 1 min of sintering at 620, 720, 820, and 920 °C, respectively; and (**f**) after 30 min of sintering at 1250 °C.

**Figure 9 materials-13-05356-f009:**
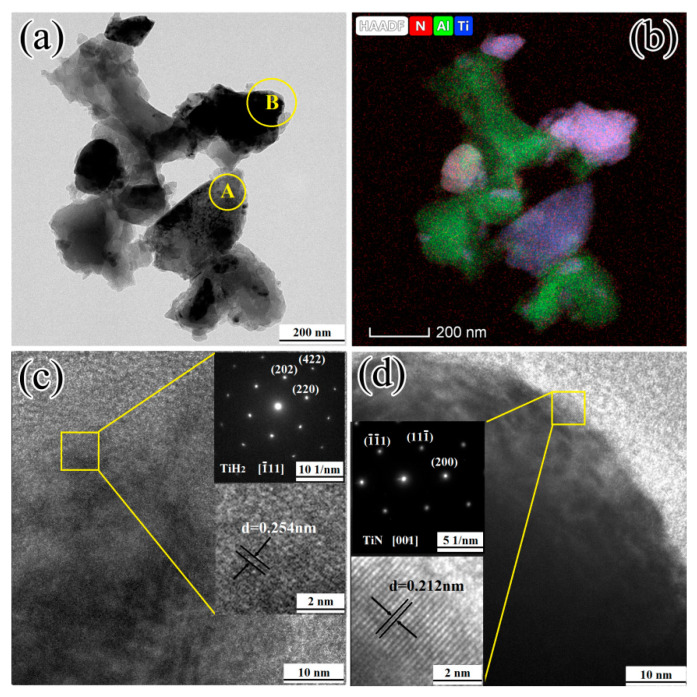
(**a**) TEM image; (**b**) elemental mapping of N, Al, and Ti; and (**c**,**d**) high-resolution transmission electron microscopy (HRTEM) image and selected area electron diffraction (SAED) pattern of 1TiH_2_/1.15Al/1TiN after one minute of sintering at 480 °C, respectively.

**Figure 10 materials-13-05356-f010:**
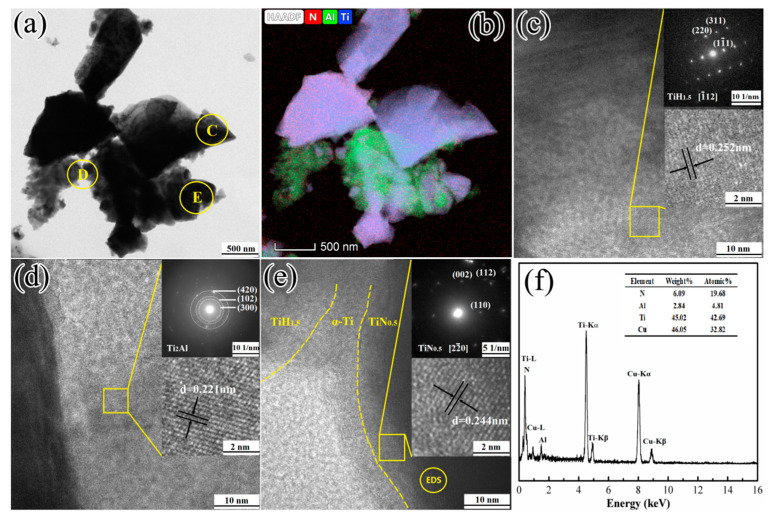
(**a**) TEM image; (**b**) elemental mapping of N, Al, and Ti; (**c**–**e**) HRTEM images and electron diffraction patterns; and (**f**) EDS analysis of the 1TiH_2_/1.15Al/1TiN powder system after one minute of sintering at 620 °C.

**Figure 11 materials-13-05356-f011:**
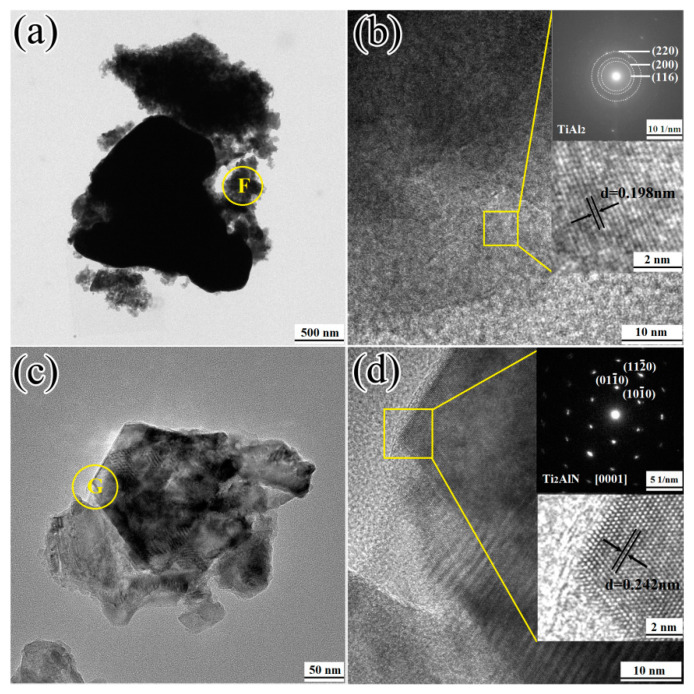
(**a**) TEM image; and (**b**) HRTEM image and electron diffraction pattern of 1TiH_2_/1.15Al/1TiN after one minute of sintering at 720 °C; (**c**) TEM image; and (**d**) HRTEM image and electron diffraction pattern of 1TiH_2_/1.15Al/1TiN after 30 min of sintering at 1250 °C.

**Table 1 materials-13-05356-t001:** Technical features of the raw materials.

Raw Materials	Manufacturer	Purity (wt %)	Particle Size
Titanium hydride (TiH_2_)	Aladdin reagent	99.0	<45 µm
Titanium (Ti)	Aladdin reagent	99.9	<45 µm
Titanium nitride (TiN)	Aladdin reagent	99.0	<4 µm
Aluminum (Al)	Aladdin reagent	99.9	<25 µm
Aluminum nitride (AlN)	Aladdin reagent	99.5	<75 µm
Tin (Sn)	Aladdin reagent	99.9	<25 µm

**Table 2 materials-13-05356-t002:** Summary of the major phase components of products following sintering under different conditions.

Sintering Temperature (°C)	Holding Time (min)	Phase Components Determined by XRD	Major Phase	The Content of the Ti_2_AlN (wt %)
Mixed powders	/	TiN, TiH_2_, Al, Sn	TiN, TiH_2_, Al	/
480 °C	1 min	TiN, TiH_2_, TiH_1.5_, Al, Sn	TiN, TiH_2_, Al	/
620 °C	1 min	TiN, TiH_1.5_, α-Ti, TiN_0.5_, Ti_2_Al, Al	TiN, TiH_1.5,_ Al	/
720 °C	1 min	TiN, Ti_2_Al, TiAl_2_, TiAl, Ti_2_AlN	TiN, Ti_2_Al, TiAl_2_	/
820 °C	1 min	TiN, TiAl, Ti_2_Al, TiAl_2_, Ti_2_AlN	TiN, Ti_2_Al, TiAl_2_	/
920 °C	1 min	TiN, TiAl, Ti_2_Al, TiAl_2_, Ti_2_AlN	TiN, TiAl	/
1150 °C	30 min	Ti_2_AlN, TiN, TiAl	TiN, Ti_2_AlN	26.68
1200 °C	30 min	Ti_2_AlN, TiN, TiAl	Ti_2_AlN	78.16
1250 °C	30 min	Ti_2_AlN, Ti_4_AlN_3_	Ti_2_AlN	96.68
1300 °C	30 min	Ti_2_AlN, Ti_4_AlN_3_, TiN, TiN_0.5_	Ti_2_AlN	76.39
1350 °C	30 min	Ti_2_AlN, Ti_4_AlN_3_, TiN, TiN_0.5_, Al_2_O_3_	Ti_2_AlN, Ti_4_AlN_3_, TiN	42.12

Note: / represents the deficiency of the corresponding data.

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
