# Peer review of "Formation Mechanism of High-Purity Ti2AlN Powders under Microwave Sintering"

_materials, 2020, doi:10.3390/ma13235356_

Round 1

Reviewer 1 Report

The authors prepared the samples Al2TiN under microwave sintering and studied it using XRD, SEM, TEM, etc. Before going to publish it, the authors have to be modified according to the following comments.

  1. Why authors used Ar atmosphere for sintering with microwaves. 
  2. Al, Ti, and N have not produced temperature under microwaves, but authors obtained 1250 oC. Need explanations with suitable references. 
  3. Hou the authors control the temperature at 1250 oC for 30, recommended putting a microwave experimental setup image in the revised manuscript.
  4. And also heating profile of with and without samples for a microwave furnace.
  5. Any relation is there for endothermic peaks in DSC and microwave absorption properties of Ti/AL/N?.
  6.  To realize the advantages of microwave heating, authors have to compare their results with conventional sintering methods.   
  7. XRD analysis, authors showing different phases of Al/Ti/N. Recommended giving suitable powder diffraction file numbers or references for that. 

Reviewer 2 Report

  • The paper is well written and the findings of the research study proves the proposed idea clearly, figures and graphs are explained well, however in the abstract it is mentioned that this method is favorable for low temperature preparation of Ti2AlN. While later the temperature has been kept higher in the study for achieving high purity of Ti2AlN power.

  • Writing in some figures and graphs are relatively small which is hard to read, I would suggest to increase its sizes (Fig1-7,9-11).

Reviewer 3 Report

The topic is of interest. The manuscript is well organized. The structure is correct as it is:

  1. Introduction
  2. Experimental procedure
    • 2.1. Materials and methods
    • 2.2. Powder characterization
  1. Results and Discussion

      3.1. DSC analysis

      3.2. Factors influencing the synthesis of Ti2AlN powders

              3.2.1. Raw material systems

              3.2.2. Aluminum content

              3.2.3. Holding time

              3.2.4. Sintering temperature

      3.3. Morphology and structure of Ti2AlN powders

      3.4. Reaction mechanism for Ti2AlN formation

              3.4.1. Reactions at temperatures lower than 660°C

              3.4.2. Reactions at 660–720°C

              3.4.3. Reactions at 720–1150°C

              3.4.4. Reactions at 1150–1350°C

  1. Conclusions (NOTE THAT IT SHOULD BE NUMBERED WITH 4.)

Here is a list of further comments for improving the manuscript:

  1. Page 1: … ternary compounds have a general chemical formula of Mn+1AXn
  2. Page 2, 1st paragraph: This phrase needs correction: "… due to the thermal explosion reaction process is complicated and the narrow stability interval of Ti2AlN at high temperature, different impurity phases, such as TiAlx and TiNx, are generally formed during synthesis [24]".
  3. Page 2: Instead of “stannum” (latin) write “tin”.
  4. Page 2: Provide information about the diameter of the agate milling spheres.
  5. Page 2: What is “an alumina crucible with no cold compaction”?
  6. Page 3: Write “Rietveld method” (instead of “rietveld method”).
  7. Page 7: Figure 8 should be located after the 1st paragraph of subsection “3.3. Morphology and structure of Ti2AlN powders”.
  8. Page 8: Figure 9 should be located after the 2nd paragraph of subsection “3.3. Morphology and structure of Ti2AlN powders”.
  9. Page 8: When mentioning “SAED” for the first time, it should be written “selected area electron diffraction (SAED)”.
  10. Throughout the whole manuscript, instead of “crystal surface exponents” it should be “Miller indices of the crystal surface” or indices of crystallographic plane.

Round 2

Reviewer 1 Report

The authors addressed all of my comments but still, they have escaped one of the queries that 'why Ar gas is used in the experiment'. The author's replay is not acceptable that 'preventing the 'oxidation' in the experiment'. Ar gas can interact with microwaves and generate plasma, That is the advantage/disadvantage of Ar gas in microwave furnace/reactor. So, I recommended to the authors rectify the errors in the experimental setup/arrangement in the manuscript. 
